# Children’s and Adolescents’ Use of Nature During the COVID-19 Pandemic in a Very Green Country

**DOI:** 10.3390/ijerph21111530

**Published:** 2024-11-18

**Authors:** Vegard Gundersen, Zander Venter, Line Camilla Wold, Berit Junker-Köhler, Sofie Kjendlie Selvaag

**Affiliations:** 1Norwegian Institute for Nature Research, 2624 Lillehammer, Norway; line.wold@nina.no (L.C.W.); berit.kohler@nina.no (B.J.-K.); sofie.selvaag@nina.no (S.K.S.); 2Norwegian Institute for Nature Research, 0855 Oslo, Norway; zander.venter@nina.no

**Keywords:** childhood, nature contact, natural environment, neighborhood, outdoor play, urban forest

## Abstract

Play, physical activity, and social interaction with other children in nature are important for healthy and social development in childhood and adolescence. The extent to which lockdown policies during the COVID-19 pandemic impacted the way children engaged in outdoor activities remains unclear, especially in a country with free access to abundant nature during the pandemic. We performed a national survey of parents (*n* = 1002) in Norway in January 2021 to uncover changes in outdoor play for children (6–12 years, *n* = 396) and adolescents (13–19 years, *n* = 606) compared with the situation before the lockdown on 12 March 2020. Ten months after the start of the COVID-19 lockdown, 38% of the parents reported that their children played and spent time outside ‘less than usual’ in their neighborhood, compared with 15% who reported ‘more than usual’ time spent outside. Parents indicated that the children’s play activities were highly organized and institutionalized, and when the activities ceased, their children had less motivation to spend time outdoors and tended to engage more in screen-based activities indoors. We conclude that while children and adolescents had many opportunities to be outdoors in natural settings during the COVID-19 pandemic restrictions, they did so much less than before the lockdown.

## 1. Introduction

For children, the positive effects of time spent in nature have beneficial mental and physical health outcomes [1,2,3,4,5]. Being in nature has the same relaxing and restorative effects on children as it does on adults, as nature is stimulating, leads to a reduction in stress, and contributes to increased cognitive concentration [1,6,7,8]. However, in recent decades, many studies have pointed to a trend of decreasing nature contact among children [9,10]. Even studies conducted before the COVID-19 pandemic found that children were less likely to play and stay in their neighborhood (within walking and cycling distance from home) and nature areas compared with children of a previous generation [11]. Reasons for this include the fact that the children’s leisure time is far more organized and institutionalized (e.g., [12]), that the parents in general had greater concerns about the traffic situation and safety in general [13,14], and that a preference for indoor screen-based activities has increased tremendously among children and adolescents (e.g., [15]). Despite growing international research on constraints for children’s nature engagement (e.g., [16,17]), there is a need for contextual studies including different sociocultural and environmental settings, and a better understanding of how family and everyday life situations affect the possibilities for children’s engagement with nature [3,4,18,19].

Numerous studies have shown an increased demand for recreational space during the COVID-19 outbreak (e.g., [20,21,22]), which remained as the pandemic progressed [23]. However, these studies largely concern the adult population. How children’s and adolescents’ outdoor activities and contact with nature changed during the pandemic is less understood. Several research papers about children confirm that the COVID-19 pandemic exacerbated physical inactivity and sedentary activities due to both reduced opportunities to engage in physical activity and to demands for maintaining physical distance during periods of lockdown [24,25]. Additionally, mental health and well-being are regarded as important [5]. These papers report on studies conducted in the USA [5,26], China [27], Croatia [28], Canada [29,30], Denmark [31], and Italy [32]. By contrast, Nathan et al. [33] did not identify any changes in total physical activity for children aged 5–9 years before and during COVID-19 in Australia, although they did identify a shift away from organized activities to more unstructured activities in different outdoor and indoor settings. Howlett and Turner [34] examined the amount of time families in the U.K. spent outside during the COVID-19 restrictions, and they identified an urban–rural difference: most rural children spent more time outdoors and most urban children spent more time indoors, which, in the former case, was probably because green spaces are more accessible in rural settings in the U.K. The above-mentioned studies primarily confirm the changes in physical activity among children due to restrictions on movement during the pandemic, but very few of the studies explored the type of changes in contact with nature regarding differing organized and unorganized play settings, and whether or not children stayed in neighborhood and nature areas when they spent time outdoors.

In Norway and the Nordic countries, children’s contact with nature is considered an important part of childhood and growing up (e.g., [35,36]). Stimulating and facilitating children to experience nature and outdoor life is also enshrined as a very important political objective in Norway [37]. Almost 98% of children and adolescents (6–19 yrs) in the Norwegian population participate in outdoor recreation at least once per year, a participation rate that has been relatively stable in recent decades [38]. However, the frequency (number of visits) of outdoor recreation for children and adolescents has been decreasing since the first survey was undertaken in 1997 [38,39]. The Nordic tradition of outdoor life (friluftsliv, meaning life in the open air) is characterized by its simplicity and popularity, and focuses on spending time outdoors in the landscape, both for general well-being and for encounters with nature that are removed from a context of formal competition [40]. Friluftsliv is enabled in Norway by the large tracts of natural areas, both privately and publicly owned, which are made accessible to the public by law (see the Outdoor Recreation Act of 1957 [41]). Norway, Sweden, and Finland together have the most peri-urban forested areas in Europe, which include forests, rivers, lakes, and coastlines, as well as intra-urban blue–green spaces, including parks and public bathing areas [42].

Norway is rich in natural landscapes and, in a national representative survey, 97% of parents of children in the age range 6–12 years reported that their children had good access to nature within walking and cycling distance from home [40]. Despite this, the parents reported that their children used nearby nature spaces far more sporadically than outdoor-developed spaces such as playgrounds and sports facilities [40]. Despite strong traditions of outdoor life and good accessibility to nearby nature, social and cultural trends seem to influence children’s engagement with nature in the same way in ‘green’ Norway as they do in many other European countries [39,40]. Studies of barriers to children’s outdoor life and nature use underline that social factors related to time pressure are more significant barriers to outdoor play than environmental factors such as accessibility, safety, and landscape quality (e.g., [12]). During the COVID-19 pandemic, rather than green areas functioning as potential sources of COVID infection, as in most other European countries [43,44], in ‘green’ Norway they were perceived as an opportunity for shelter, recreation, and children’s stay and play in a manner that ensured social distancing [22,23].

In contrast to outdoor spaces, restrictions on predominantly indoor institutions were imposed for children and adolescents when the COVID-19 lockdown started in Norway on 12 March 2020. The measures included the closure of schools and preschools, indoor sports and training facilities, and all types of organized leisure and physical activities [22]. On 7 April 2020, there was a first partial lifting of measures that included the reopening of preschools, followed by the reopening of schools for Grades 1–4 on 17 April. On 1 June, there was a final lifting of measures, including the reopening of all schools, and on 29 July, the reopening of all organized activities. During the second wave of COVID-19 infection, from September 2020 to January 2021, national measures were replaced by local restrictions as needed, and in January 2021, by the time our survey was distributed, the situation for children and adolescents was largely normalized following a decreasing transmission trend. During the reopening period of preschools and schools, there were, to some extent, adaptations to teaching in smaller groups in addition to more teaching outdoors. Accordingly, we conducted a national representative questionnaire survey that opened on 12 January 2021, targeting parents with children and adolescents aged 6–19 years. Our data represent the ten-month period after the initial phase of the COVID-19 pandemic lockdown (12 March 2020).

In this context of changing interaction between children and nature in a country with abundant access to nature areas in the neighborhood, it is interesting to study the effects of the initial phase of the COVID-19 lockdown from 12 March 2020 to January 2021. ‘Neighbourhood’ was defined as including a wide range of different environments, such as private gardens, abandoned areas, playgrounds, streets, forests, and parks within walking and cycling distance from home. All these categories included, to differing extents, nature elements and structures such as trees and vegetation. ‘Natural areas’ were defined as including natural vegetation either within or outside a city boundary and consisted of a wide range of both blue and green environments [12,40,42]. The main research questions we aim to answer are: During the COVID-19 pandemic, (1) how did time outdoors change for children and adolescents? (2) Did the type of outdoor setting (natural versus neighborhood) matter? Lastly, (3) how did observed changes correlate with social, environmental, and demographic factors?

## 2. Materials and Methods

### 2.1. Target Population, Sampling Technique, and Sample

The survey was administered during the period 12 to 30 January 2021 to a nationwide panel through the polling company Norstat. Norstat panel participants are recruited through a variety of online and offline methods, with more than 50% recruited via direct personal contact by telephone. The panelists consent to participate in survey-based research when they register to join the panel and get payment in return for each survey they answer. This survey was web-based, and respondents answered directly via mobile phone, tablet, or PC. To secure enough evaluations for each question, we aimed to collect 1000 responses, stratified at the county level to include all 15 counties in Norway. Norstat invited a total of 4105 parents and other types of caregivers (hereafter referred to collectively as ‘parents’) with children and adolescents within the age range 6–19 years to participate. The respondents were initially encountered via a screening question asking whether or not they were a parent of children/adolescents aged 6–19 years, and subsequently, 1244 respondents submitted the questionnaire form. In total, 111 respondents were screened out because their children were outside the age range and a further 131 respondents did not complete the questionnaire in full, resulting in a final sample of 1002 respondents representing our target population.

Because the respondents were recruited from a survey panel consisting of volunteers, they are continuously tested by the polling company to ensure they are representative of the general population of Norway. There is no standardized method for calculating a response rate in this type of survey [45,46]. We considered the sample of respondents representative of the target population with respect to the children’s gender, parent’s age, geographical distribution in six regions, and family structure (child/adolescent lived in one home vs. more than one home) among the target population of Norwegian children and adolescents aged 6–19 years (Table 1). To avoid over-representation of parents with children/adolescents engaged in outdoor recreation activities, the respondents were initially informed that the survey concerned children’s/adolescents’ leisure time in general. We considered the sample to be valid in terms of reaching both children/adolescents who were currently engaged in nature-based activities, had previously engaged in nature-based activities, and had never engaged in nature-based activities. The respondents were asked to answer the questionnaire on behalf of their oldest child within the specified age range (6–19 years). This resulted in a slight overweight of parents representing adolescents in the upper age groups (especially 16–17 years, which was approximately 5% higher in the sample compared with the target population) compared with children in the lower age groups (especially 7–9 years, approximately 2.8% lower in the sample compared with the target population). In our sample, the median annual household income (813,300 NOK) was higher than in the target population (590,400 NOK [47]). As is common in such national surveys of parents [40], the proportion of immigrant parents was lower (10.2%) in the sample than in the official national statistics relating to the Norwegian population (17.0%). However, the number of immigrants who participated in our web-based survey was much higher than expected, based on our reading of earlier studies (e.g., [39]).

### 2.2. Questionnaire and Measurement Methods

The design of the questionnaire was based on a similar survey targeting children’s and adolescents’ use of nature in Norway conducted by Skar et al. [39]. Content in the survey was used to assess, for example, the location(s) of children’s and adolescents’ leisure time, parent/child demographics, a change in the child’s leisure behaviors and use of nature, as well as the parent’s evaluation of the consequences for their child of the suspension of organized activities during the COVID-19 pandemic (Appendix A). The selected items, including the measurement scale used in this secondary analysis, are listed in Table 2.

We asked the parents about their child’s play and time spent outdoors ten months after the start of the pandemic lockdown (January 2021) in two different settings, namely in the neighborhood and in natural areas (Table 2). Responses were reported in three categories: ‘less than usual’ (score 1), ‘as usual’ (score 2), and ‘more than usual’ (score 3). The category ‘as usual’ was set as a baseline and defined as the year 2019 before the COVID-19 pandemic evolved in 2020. We provided these definitions as a short text in the introduction to the questionnaire. The parents were subsequently asked to compare their child’s leisure behavior before and during the pandemic. Responses to seven statements were reported using a 7-point Likert-type scale, ranging from ‘completely disagree’ (score 1) to ‘completely agree’ (score 7) (Table 2). Based on previous studies of children’s leisure time [12,36,39], we asked the parents how their child or adolescent had experienced the suspension of organized outdoor activities immediately after the initial phase of lockdown and their possible motivation for changed behavior in terms of time spent outdoors in a new setting. For this question, we provided six statements measured by a seven-point Likert scale ranging from ‘completely disagree’ (score 1) to ‘completely agree’ (score 7) (Table 2). In addition, we asked the parents to answer key demographic, social, and environmental questions (Table 2). The definitions of these standard background variables relating to parents and children followed official definitions for statistics in Norway [38,47], and they have been used in several similar national surveys targeting children and adolescents (e.g., [40,48]).

### 2.3. Data Processing and Analyses

After data collection was complete, a dataset (.csv file) was received from Norstat. Further data cleaning and verification were completed by us (e.g., target criteria, incomplete answers). The data were analyzed using IBM’s SPSS Statistics 27. Overall, descriptive statistics were calculated as means (standard deviation) and percentages (%) for all variables and were presented for selected variables (Table 2). A Pearson Chi-square test was used to assess differences between categorical variables (Appendix A). For this, categorical data for selected demographic, social, and environmental variables were collapsed into two levels. Associations between changed behavior and demographic, social, and environmental factors were assessed using a Pearson point biserial correlation (see similar analyses used in [24]). Statistical significance was set at *p* < 0.01. Analyses were completed and checked by two of the authors.

### 2.4. Ethics Approval

The data used in the analyses were collected in line with ethical standards. The respondents, who had given prior consent to participate in research, were drawn from a panel by a third-party company, Norstat. Norstat has strict requirements for privacy and data security, and works in line with both the Norwegian Data Protection Authority’s guidelines and the provisions of the Personal Data Act [49]. Collection and analyses of data were carried out with approval from the Norwegian Centre for Research Data (NSD). The data file provided to us did not include any identifiable personal information and thus the participants (*n* = 1002) were fully anonymized.

## 3. Results

### 3.1. Children’s and Adolescents’ Use of Nature Before and During the COVID-19 Pandemic

Descriptive statistics of statements and key demographic factors for the parents and children are provided in Table 3. Of the 1002 parents who completed the survey, the majority were female (52.1%), born in Norway (89.8%), had one home (79.5%), were either married or cohabiting (87.4%), and had an annual household income of >1,000,000 NOK (50.4%). Ten months after the initial lockdown period, most of the parents reported no change in their children’s time use of neighborhood and natural areas compared with the corresponding times for the 2019 baseline (Figure 1). However, in this context, it is more interesting to look at those parents who reported that their children played less (38%) and spent less time (28%) in the neighborhood and natural areas. By contrast, far fewer parents reported that their children had increased their use of their neighborhood (15%) and natural areas (18%). When considered as a net change, the parents reported that during the pandemic, 23% of the children played and spent less time in their neighborhood, and 10% played and spent less time in natural areas.

Overall, many of the demographic, social, and environmental variables were significant (*p* < 0.05) with regard to children’s and adolescents’ use of both their neighborhood (child age, rural vs. urban living, number of homes lived in, parent’s gender, parent’s age, annual household income, ethnicity, caregiver, availability of nearby nature) and natural areas (geographical region, rural vs. urban living, parent’s gender, parent’s age, ethnicity, availability of nearby nature) during the COVID-19 pandemic (Appendix A). Details of all associations between perceived changes in children’s use of their neighborhood and natural areas during the pandemic, as well as demographic, social, and environmental factors, are provided in Table 4. The following selected associations from the Pearson point biserial correlation (r > 0.1, *p* < 0.01) are highlighted: compared with children (6–12 years), adolescents (13–19 years) were associated with less use of their neighborhood (−0.14) and natural areas (−0.20). Also, children and adolescents who had good access to natural areas were associated with more use of their neighborhood (0.10) and natural areas (0.12) than those who had poor access.

### 3.2. Parent’s Evaluation of Children’s and Adolescents’ Use of Leisure Time During the COVID-19 Pandemic

Details of the associations between perceived changes in children’s and adolescents’ use of leisure time, as well as demographic, social, and environmental factors, are provided in Table 4. The following selected associations from the Pearson point biserial correlation (r > 0.1, *p* < 0.01) are highlighted. Adolescents (13–19 years) spent more time on screens and social media (0.17) than children (6–12 years). In addition, older parents (40–80 years) had adolescents who spent slightly more time on screens and social media than younger parents (20–39 years) (0.10). Also, family structure was associated with time spent on screens and social media in that the children and adolescents who lived in two or more homes spent more time on screens than those who had only one home (0.10). The same variable, family structure, was associated with the variable ‘less time with friends’, as children and adolescents who had two or more homes spent less time with friends than those who had only one home. Some of the children and adolescents stopped engaging in organized leisure activities during the COVID-19 pandemic, and children and adolescents who had good access to natural areas from their home stopped engaging in organized activities to a lesser extent than those who had poor or medium access (−0.13). The same variable, ‘stopped doing organized activities’, was also associated with geography. For example, children and adolescents who live in the capital region Oslo (0.21), and those who had parents born outside Norway (0.13), stopped engaging in organized activities to a greater extent than children who lived outside the Oslo region and had Norwegian-born parents. The variable ‘engaged in new use of the neighbourhood’ showed some significant patterns. Children (6–12 years) were more engaged in their neighborhood than were adolescents during the pandemic (−0.16), and this result was supported by a similar pattern for children and adolescents with parents in the younger age group (20–39 years) (−0.11).

### 3.3. Parent’s Evaluation of the Consequences of Suspension of Organized Leisure Activities

The suspension of organized leisure activities during the COVID-19 pandemic meant more leisure time for the children and adolescents, and provided new opportunities for social engagement, physical activity, and doing different leisure activities. The association between six statements measuring their allocations of tasks and the demographic, social, and environmental variables is summarized in Table 5. Regarding the statements that most parents agreed with, namely ‘missed meeting friends at organized activities’ (mean 5.22) and ‘missed physical activity in organized activities’ (mean 4.79), this was most prominent for those who were members of the respective organizations. We did not identify any other significant patterns for the two statements. For children and adolescents who were ‘happy to have more time on screens and social media’ (mean 4.55), we identified a significant association, namely that the boys were happier than the girls, and that adolescents (13–19 years) were happier than the children (6–12 years). The variable ‘happy to have more leisure time outdoors’ received the lowest score based on the parents’ evaluations (mean 3.60), and adolescents (13–19 years) (−0.12) whose parents were in the older age group (40–80 years) (−0.13), were not a member of organized recreation activities (−0.20), and families with two or more children (0.12) were least happy with the statement. For the statement ‘happy to spend more time with the family’, we identified significant differences regarding parents’ age and number of children, and children and adolescents whose parents were in the younger age group (20–39 years) (−0.18). We found that children and adolescents in families with two or more children (0.11) were more happy to spend leisure time with their family than families with sole children and adolescents.

## 4. Discussion

The purpose of our study was to highlight the changes in children’s and adolescents’ use of their neighborhood and natural areas during the initial phase of the COVID-19 pandemic lockdown in Norway. We found that children and adolescents played less and spent less time in those environments during the ten months after the start of the lockdown compared with the baseline before the restrictions were imposed. The largest reported change in behavior related to more leisure screen-based activities and correspondingly fewer outdoor activities. These findings confirm that pandemic-related restrictions were unfavorably related to children’s and adolescents’ more limited use of life outdoors [24,50,51], even in a country with free access to abundant nature during the COVID-19 pandemic. It is well known that spending time outdoors is related to play opportunities, physical activity, less sedentary time, less stress, and a number of other benefits such as improved mental health [2,3,5,52]. Regarding physical activity, indoor play-based activities do not seem to replace active play outdoors [24]. Our results corroborate a well-established downward trend in play and time spent in nature for children and adolescents in Norway and elsewhere in the Western world [9,12]. Moreover, Norway is a country rich in nature and most children have ready access to nature for outdoor play and exercise [40]. Our results should contribute to the current field of knowledge because previous research has generally focused on decreasing access to nature and perceived decreasing attractiveness of nature as the main hindrances to children’s engagement with nature [18]. Less literature is concerned with how children’s and adolescents’ socio-ecological everyday settings (time pressure, indoor activities, social media, institutionalized and organized activities) affect their engagement [12,53]. In the following sections, we discuss our general findings regarding changes in children’s and adolescents’ behavior and use of leisure time during the first ten months of the COVID-19 pandemic in Norway, in addition to the main demographic, social, and environmental settings associated with their leisure time.

### 4.1. Changes in Behavior and Use of Leisure Time

During the lockdown, the leisure situation for children and adolescents changed, and especially the suspension of organized activities gave families opportunities to allocate their time to other tasks. Our data indicated that children and adolescents did not use their freed-up time to play and be outdoors but rather increased the time they spent on screen-related indoor activities. According to Moore et al. [24], families reconnected through leisure, although it seems that most often this was primarily through sedentary leisure activities and not through new hobbies and activities. A study before the pandemic from Norway identified that the main barriers to increasing children’s engagement with nature was that their available leisure time was taken up with organized sports and leisure activities, and homework [12]. According to our results, based on parents’ evaluations, although children and adolescents had less busy weekdays during the lockdown, it did not seem to lead to increased engagement in outdoor activities. In the past, the neighborhood and nearby natural areas were important informal meeting places for children’s games and play, without the adults necessarily being present [8,11,36]. Regarding the risk of COVID-19 infection and the need for social distancing, undoubtedly parents were concerned about their children’s contact with other children outdoors [24,25]. The parents thus faced a new concern that was more difficult to cope with than the traffic and safety situation in general, which were noted as important concerns that hindered children’s time spent outdoors in near-home contexts before the COVID-19 pandemic [12,13]. Ten months after the initial phase of the lockdown (i.e., at the time when the parents answered the survey), the infection rate was decreasing and the situation for the children was normalized. This indicates that the infection situation was not the only reason for less time spent outdoors, and, in other studies, this is explained by the fact that the children had been more used to indoor activities, mostly increased screen-based activities and spending more time on social media [24,27,33]. Schmidt et al. [25] explain parents’ concerns about their children’s time spent outdoors as an adaptation to the risk of infection that their children were exposed to. It is difficult to eliminate the risk of infection outdoors when children play and have close interaction with other children. By allowing more screen time, parents gained better control over their children indoors. Another strategy used by parents was either to keep the children indoors at home or to take them out into nature only under adult supervision [24]. Even though adolescents most often are outside without adults present, our data revealed that the adolescents (12–19 years) in general preferred outdoor activities to a lesser extent than did the children (6–12 years). Other studies explain the decline in outdoor activity by the fact that there was a change in attitude and mentality during the COVID-19 pandemic [5], and indoor activities gained far greater legitimacy among parents [50]. Furthermore, it has been shown in other studies that such attitudes can be difficult to change and that it will take a long time to return to the situation before the pandemic if that is the desired goal (e.g., [54]). To corroborate these findings, further research is needed to evaluate the post-COVID-19 situation in terms of time spent outdoors, and especially the level of play in natural settings among children.

In a separate study, children themselves stated that screen time was their ‘free time’ in a busy everyday setting that included homework, sports, and other activities [12]. The situation in ‘green’ Norway seems to be much the same as in other Western countries, where spending time outdoors in natural settings may be at risk of being forgotten among the scheduled targets and competing indoor screen activities (e.g., [9,10,53]). Playing games also include landscapes and natural scenery on the screen, but these settings are perfectly arranged for the game activity and cannot replace physical nature experiences [55]. We identified that 87% of the children had either good or very good access to nature, and the availability of nature is significant for children’s and adolescents’ time spent outdoors. In a similar study conducted among parents in Norway, 88% stated that that their child had good or very good opportunities for play and time spent outdoors in nearby nature, but that those areas included only sporadic daily use by children compared with developed spaces such as sports and recreational facilities [40]. In this respect, a question that arose is whether the availability of natural areas is a relevant barrier to most children’s time spent outdoors in Norway or whether the parents’ evaluations were based on a perception that access to nature spaces could always be improved in their neighborhood?

Our data confirmed the importance of organized activities for children’s life outdoors in Norway compared with other Western countries (e.g., [24]), where only 7% of the parents reported that their children or adolescents did not participate in any kind of organized leisure activities. During the lockdown period, all kinds of organized activities were temporarily closed, and for most children, this entailed crucial behavioral and social changes [24,31]. The statement that their child did not start a new hobby/activity during the pandemic got a low score (mean: 2.49). This can be interpreted as meaning that there were few opportunities for children to explore new arenas during the pandemic, and this was especially the case for adolescents. We noted that parents considered that their child had less busy weekdays and spent more time with adults, and these findings underline that parents’ motivation to engage in recreational activities with their children is crucial for encouraging outdoor play, especially unsupervised play [12,16,24].

### 4.2. Demographic, Social, and Environmental Settings

The most prominent demographic pattern in the survey was that children were more dedicated to spending time outdoors than adolescents and having young parents (age group 20–39 years) was significant in this respect. The parents expressed that their children were happy to have more time to spend on screen-based activities, and correspondingly less time for playing outdoors, and this change was most prominent for boys, adolescents (13–19 years), those with older parents (40–80 years), and those who lived in two or more homes. The age of adolescents (13–19 years) and parents (40–80 years) is therefore correlated. Children (6–12 years) experienced less change in behavior than adolescents, probably because they had greater preferences for spending time on outdoor activities and play [24]. The observed relations to parents’ age are consistent with typical age-related declines in physical activity among the adult population [56]. Our data indicated a strong preference for indoor activities among children. In the wider literature, girls are shown to be less active outdoors than boys (e.g., [24]); however, our results showed that the boys were most happy to spend their leisure time on indoor screen-based activities.

Family situation was important for possibilities to maintain outdoor activities during the COVID-19 pandemic [24]. Our study results confirm a pattern that families that were largely engaged in outdoor recreation before the pandemic had new opportunities during the pandemic to sustain or increase their outdoor activity levels. In situations where parents had a home office and the youngest children either stayed at home or came home early from school, and the family had more leisure time in general, the family had more flexibility to engage in unstructured outdoor activities in their neighborhood or in nearby nature, as observed in a study conducted in Australia [33]. However, for most children, the closure of all kinds of organized activities during the initial phase of the lockdown had strong negative consequences for the children’s possibilities for meeting friends, engaging in physical activities, and being outdoors in general. For such children, outdoor activities were largely organized and institutionalized, and took place under the auspices of schools, after-school arrangements, or various forms of organized leisure-time activities [12,19,36].

### 4.3. Strengths, Limitations, and Future Directions

As there is still little knowledge about the effects the COVID-19 outbreak had on children’s opportunities to engage in outdoor activities, especially in natural settings, our study will contribute to filling that gap. A retrospective cross-sectional survey has obvious limitations but given the fast appearance of the pandemic and the situation that parents and children were exposed to during it, we believe that the parents’ recollections’ on the situation before and after the pandemic are a relevant indication of children’s leisure behavior. To overcome these limitations, we designed a methodology that involved asking for parents’ evaluations on children’s outdoor life without mention of COVID-19. This then gave us an indication of the main patterns of change that may be important to uncover in more detail in a cohort study. The survey questionnaire sent out in January 2021 referred to children’s use of outdoor activities retrospectively during the COVID-19 pandemic. It is unlikely, therefore, that low outdoor activity among children in the time parents answered the questionnaire influenced their answers. Our sample was a nationally representative sample of 1002 parents of school-age children and adolescents. Our study was limited by categorical data and descriptive statistics, and both the category ‘neighbourhood and nearby natural areas’ and the definitions of ‘neighbourhood’ and ‘natural areas’ were wide and included many different environmental settings. A more detailed study of the environments in their nearby settings might have led to other conclusions, such as the use of playgrounds versus more unstructured natural settings [11]. Asking parents about their children probably leads to a desirability bias, and thus might have affected our findings. There is reason to assume that parents exaggerated their children’s outdoor play in the survey (as in recreation in general, see [57]), in which case it would serve to reinforce the conclusion that fewer children played outside during the COVID-19 pandemic period. Moreover, we did not include data that explained the children’s and family’s context, such as parent’s work status, children’s play behavior, or whether parents were working from home or had lost their job due to the pandemic. Our sample of the target population was quite representative with regard to demographic, geographical, and family structure, but it underestimated the immigrant group (10.2% versus 17%). Despite this, the number of immigrants who participated in the web-based survey was much higher than in similar previous nationwide surveys (e.g., [40]). We can assume, therefore, that this bias influenced our results because immigrants generally have fewer opportunities to access natural areas and likewise tend to use them much less compared with the majority of the population [15,58,59]. Our results may not be generalizable to other countries outside the Nordic context due to the large amount of nature in the country, the right of common access to all land, and other socio-cultural settings including a different economy, demography, and family structure [40]. Future studies should explore further structural and social barriers to explain why children’s time spent in play outdoors and in nature has been declining over time and should explore the consequences of these emerging changes for children and society in general (e.g., [8,60]). There is a need for research exploring the impacts of shifting time outdoors on children’s mental health, including potential mechanisms that might be driving it [5].

## 5. Conclusions

This study adds to the growing knowledge base on the effects the COVID-19 outbreak (and associated lockdown restrictions) had on children’s and adolescents’ behavior related to play and time spent outdoors in their neighborhoods and in nature in Norway. In ‘green’ Norway, we identified a decline in children’s and adolescents’ outdoor play and time spent outdoors ten months after the start of the pandemic. This was likely due to a heavy reliance on organized outdoor activities for children, and that the closure of organized activities during the lockdown was substituted with indoor leisure activities. Our discussion outlines massive challenges in society to change children’s behavior and to let the children come out and play again. The family situations during the pandemic may have established continuing habits and preferences for indoor activities, particularly screen-based activities, which can make it even more difficult to reverse an alarming negative long-term trend in children’s time spent playing outdoors.

## Figures and Tables

**Figure 1 ijerph-21-01530-f001:**
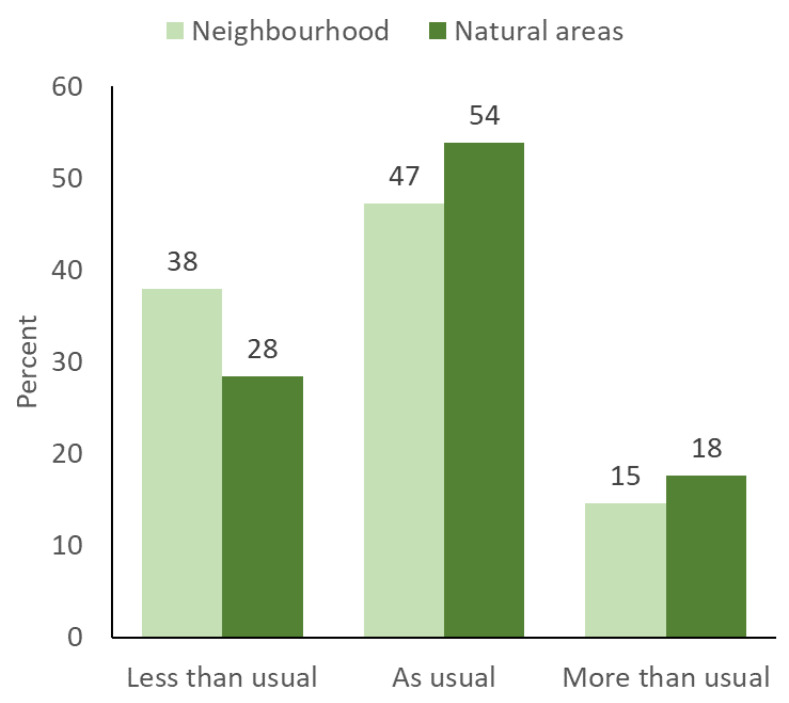
Time spent outdoors in the neighborhood and natural areas by children and adolescents ten months after the initial phase of the COVID-19 pandemic lockdown in Norway (*n* = 1002). Pearson Chi-square = 710.34, df = 4, *p* < 0.01.

**Table 1 ijerph-21-01530-t001:** Representativity of the sample compared with the target population (Statistics Norway 2021 [38]).

Survey Representativity			
	Target Population (%)	Survey Sample (%)	Difference (%)
Gender children, girls (6–19 years)	48.6	46.3	−2.3
Age children, 6 years	7.0	6.1	−0.9
Age children, 7 years	7.1	4.2	−2.9
Age children, 8 years	7.3	4.7	−2.6
Age children, 9 years	7.4	4.6	−2.8
Age children, 10 years	7.3	6.2	−1.1
Age children, 11 years	7.1	6.0	−1.1
Age children, 12 years	7.2	7.6	0.4
Age children, 13 years	7.0	7.9	0.9
Age children, 14 years	7.0	8.1	1.1
Age children, 15 years	7.0	8.0	1.0
Age children, 16 years	6.9	11.3	4.4
Age children, 17 years	7.0	13.4	6.4
Age children, 18 years	7.4	8.2	0,8
Age children, 19 years	7.4	3.9	−3.5
Region, Northern Norway (Finnmark, Troms, Nordland)	8.9	9.6	0.7
Region, Central Norway (Trøndelag)	13.8	14.4	0.6
Region, Western Norway (Møre og Romsdal, Sogn og Fjordane, Hordland, Rogaland)	22.0	20.0	−2.0
Region, Eastern Norway (Oppland, Hedmark, Akershus)	30.7	30.0	−0.7
Region, Southern Norway (Telemark, Buskerud, Aust-Agder, Vest-Agder, Vestfold, Østfold)	14.0	12.9	−1.1
Region, Oslo (Oslo)	10.6	13.2	2.6
Gender parents, women (22–77 years)	50.3	52.1	1.8
Foreign country of birth, parents ^1^	17.0	10.2	6.8
Family structure, one home ^2^	76.7	77.9	1.2

^1^ Statistics Norway’s definition of immigrants is somewhat wider (includes grandparents) than the definition used in the survey, and the difference is somewhat less. ^2^ Including children 0–17 years (children’s families, Statistics Norway 2021).

**Table 2 ijerph-21-01530-t002:** Selected parameters used in the analyses of children’s and adolescents’ behavior during the initial phase of lockdown in the COVID-19 pandemic.

Survey Details	Response Options
**Questions about play and time spent in nature**	
What is the child’s and adolescent’s everyday life like during the pandemic at the present time?	They play and stay outdoors in the neighborhood [dropdown]: - Less than before the pandemic - As usual - More than before the pandemic They play and stay in natural areas [dropdown]: - Less than before the pandemic - As usual - More than before the pandemic
**Demographic characteristics**	
**Parent:** Age and gender, Income, Sole parent, Ethnicity, Postcode, Family structure, Number of children, Employment status	[dropdown, age-continues] [dropdown, gender] [dropdown, income, 16 categories] [dropdown, sole parent, yes/no] [dropdown, ethnicity, 7 categories] [dropdown, family structure, 3 categories] [dropdown, number of children, specify] [dropdown, rural vs. urban living, 4 categories] [dropdown, work position, 12 categories]
**Child:** Age and gender Geographical region Rural vs. urban living Member of organized outdoor recreation activities	[dropdown, age-continues] [dropdown, gender] [dropdown, county, 19 categories] [dropdown, postal code, specify] [dropdown, organized activities, yes-no, specify]
**Changes in children’s leisure activities**	
Have you noticed anything different today compared to before the first COVID-19 lockdown in terms of your child’s leisure-time activities?	[dropdown, Likert scale 1–7, 1 = completely disagree, 7 = completely agree] - Spend more time on screens and social media - Spend less time with friends - Spend more time with adults - Stopped doing leisure activities - Less busy weekdays - Use outdoor areas in a different way - Started with new hobby/leisure activities
**Consequences of suspension of organized leisure activities**	
What do you think your child felt about organized activities being closed down during the COVID-19 lockdown?	[dropdown, Likert scale 1–7, 1 = completely disagree, 7 = completely agree] - Missed meeting friends at organized activities - Missed physical activity in organized activities - Happy to have more leisure time in general - Happy to have more leisure time outdoors - Happy to spend more time with family - Happy to have more time on screens and social media

**Table 3 ijerph-21-01530-t003:** Details of parents’, children’s, and adolescents’ characteristics and measured statements (*n* = 1002).

Measurements	Descriptive Statistics
**Parents’ demographic profile**	
Age, Mean (SD)	45.02 (8.02)
Gender female, *n* (%)	522 (52.1)
Annual household income, *n* (%)	
<499,000 NOK	91 (10.5)
500,000 to 999,000 NOK	338 (39.1)
1,000,000 to 1,499,000 NOK	338 (39.1)
>1,500,000 NOK	98 (11.3)
Sole parent, *n* (%)	126 (12.6)
Ethnicity, non-Norwegian, *n* (%)	102 (10.2)
Family structure, one home, *n* (%)	781 (79.5)
Number of children, Mean (SD)	2.12 (0.97)
**Children’s and adolescents’ demographic profile**	
Age, Mean (SD)/*n* (%):	
Children 6–12 years	9.24 (2.10), 394 (39.3)
Adolescents 13–19 years	15.89 (1.78), 608 (60.7)
Gender female, *n* (%)	464 (46.3)
Geographical region:	
Northern Norway	96 (9.6)
Central Norway (i.e., Trøndelag)	144 (14.4)
Western Norway	200 (20.0)
Eastern Norway	301 (30.0)
Southern Norway (incl. Telemark og Vestfold)	129 (12.9)
Oslo	132 (13.2)
Rural vs. urban living, *n* (%):	
City (>50,000 inhabitants)	393 (39.2)
Town (5000–50,000 inhabitants)	297 (29.6)
Village (200–5000 inhabitants)	217 (21.7)
Rural (<200 inhabitants)	95 (9.5)
Member of organized outdoor recreation activities, yes *n* (%)	532 (53.1)
**Changes in children’s leisure activities, Mean (SD)**	
Spent more time on screens and social media	4.82 (1.82)
Spent less time with friends	4.70 (1.89)
Spent more time with adults	4.53 (1.71)
Stopped doing organized leisure activities	2.68 (2.10)
Less busy weekdays	4.56 (1.71)
Engaged in new use of the neighborhood	3.16 (1.74)
Started new hobby/leisure activities	2.49 (1.82)
**Consequences for children of suspension of organized leisure activities**	
Missed meeting friends during organized activities	5.22 (1.90)
Missed physical activity during organized activities	4.79 (2.04)
Happy to have more leisure time in general	4.10 (1.73)
Happy to have more leisure time outdoors	3.60 (1.59)
Happy to spend more time with family	4.22 (1.58)
Happy to have more time to spend on screens and social media	4.55 (1.81)

**Table 4 ijerph-21-01530-t004:** Associations between perceived changes in children’s and adolescents’ behavior, and demographic, social, and environmental factors. The selected associations from the Pearson point biserial correlation (r > 0.1, *p* < 0.01) are highlighted in green.

	Play and Stay in the Neighborhood	Play and Stay in Natural Areas	More Time on Screens and Social Media	Less Time with Friends	More Time with Adults	Stopped Doing Leisure Activities	Less Busy Weekdays	Engage in New Use of the Neighborhood	Started New Hobby/Leisure Activities
Child’s gender (boy, girl)	0.027	−0.003	−0.200	−0.012	0.005	<0.001	−0.006	0.039	−0.032
Child’s age (6–12, 13–19)	−0.135 *	−0.200 *	0.171 *	0.027	−0.036	−0.033	0.046	−0.159 *	−0.088
Child is a member of organized recreation activities (yes, no)	−0.084 *	−0.077	−0.042	−0.057	−0.007	−0.007	0.059	−0.067	−0.027
Parent’s age (20–39, 40–80)	−0.071	−0.098 *	0.101 *	0.036	−0.019	0.040	0.034	−0.110 *	−0.063
Parent’s annual household income (<800,000, >800,000)	0.085	0.083 *	−0.006	0.021	0.035	−0.003	0.052	−0.012	−0.056
Sole parent (yes, no)	0.024	0.038	0.003	−0.031	−0.018	−0.054	−0.009	−0.004	−0.058
Family structure (one home, two or more homes)	−0.073	−0.063	0.102 *	0.108 *	0.041	0.060	0.014	0.035	0.019
Number of children (one child, two or more children)	0.049	0.085 *	−0.018	−0.039	−0.005	−0.014	0.032	0.084 *	−0.003
Availability of natural areas (poor, medium, good)	0.101 *	0.121 *	−0.060	<0.001	−0.002	−0.125 *	0.044	0.048	0.011
Rural vs. urban living (urban > 5000, rural < 5000)	−0.004	0.031	−0.021	0.033	−0.012	−0.041	0.002	−0.022	−0.019
Geography (Norway without capital city region, capital city region)	−0.005	−0.030	0.069	0.047	0.097 *	0.208 *	0.060	0.023	−0.021
Geography (Norway-born, foreign-born)	−0.030	−0.039	0.030	−0.018	0.008	0.125 *	−0.039	0.004	0.062

* *p* < 0.01.

**Table 5 ijerph-21-01530-t005:** Associations between perceived effects of organized activities during lockdown and demographic, social, and environmental factors assessed using a Pearson point biserial correlation. The selected associations from the Pearson point biserial correlation (r > 0.1, *p* < 0.01) are highlighted in green.

	Missed Meeting Friends at Organized Activities	Missed Physical Activity in Organized Activities	Happy to Have More Leisure Time in General	Happy to Have More Leisure Time Outdoors	Happy to Spend More Time with Family	Happy to Have More Time on Screens and Social Media
Child’s gender (boy, girl)	0.047	0.008	−0.083	−0.037	0.009	−0.172 *
Child’s age (6–12, 13–19)	0.040	0.029	0.060	−0.119 *	−0.073	0.134 *
Child is a member of organized recreation activities (yes, no)	−0.129 *	−0.120 *	−0.096 *	−0.196 *	−0.096 *	−0.034
Parent’s age (20–39, 40–80)	−0.024	−0.027	−0.074	−0.127 *	−0.183 *	−0.055
Parent’s annual household income (<800,000, >800,000)	0.005	<0.001	0.069	−0.009	<0.001	−0.039
Sole parent (yes, no)	−0.006	−0.019	0.056	0.001	−0.006	0.044
Family structure (one home, two or more homes)	0.010	0.035	−0.066	−0.039	−0.083	0.005
Number of children (one child, two or more children)	−0.009	−0.016	0.065	0.116 *	0.105 *	0.022
Availability of natural areas (poor, medium, good)	−0.032	−0.065	0.021	0.057	0.003	−0.042
Rural vs. urban living (urban > 5000, rural < 5000)	−0.068	−0.060	0.032	0.017	0.004	0.005
Geography (Norway without capital city region, capital city region)	0.018	0.044	0.028	−0.017	−0.008	0.020
Geography (Norway-born, foreign-born)	0.042	0.060	−0.001	0.036	0.032	0.052

* *p* < 0.01.

## Data Availability

The data collected for this study may be available on request from the corresponding author.

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
