# Peer review of "Children’s and Adolescents’ Use of Nature During the COVID-19 Pandemic in a Very Green Country"

_ijerph, 2024, doi:10.3390/ijerph21111530_

Round 1
Reviewer 1 Report
Comments and Suggestions for Authors
- A brief summary The main aim of this paper was to investigate whether children spent more or less time outdoors and the factors that influenced any changes.
- General concept comments
Article: The article was a well conceived and executed study that is well described, including limitations of the study and data. The conclusions are well supported by the results provided.
Review: The article positioned the research explicitly against other studies in this field. The article referred to most of the major literature available in the discussion topic relevant to the study. - Specific comments No inaccuracies could be found in areas such as table referencing, etc. Final manuscript will need to amend (Author, et al. 2016), remove sentence space in line 112 and 215 and ensure all references are formatted correctly. MDPI advise that this work will be done by them.
- The manuscript is clear, relevant for the field and presented in a well-structured manner
- The references were mostly recent and were appropriate.
- The manuscript is scientifically sound and the experimental design is appropriate to test the hypothesis but with acknowledged limitations of self-report design.
- A similar study could be constructed based on the methodological design and results reporting. The data is presented clearly and explicitly with appropriate selection of total data in relation to the research question.
- The conclusions consistent with the evidence and arguments are presented
- As the data and study was funded by a reputable government body and data collection carried out by a government sanctioned organisation it his highly unlikely that there are any ethical issues. However, an ethics approval number would further give confidence and and allow easier auditing.
Author Response
Reviewer 1
Open Review
( ) I would not like to sign my review report
(x) I would like to sign my review report
Quality of English Language
(x) The quality of English does not limit my understanding of the research.
( ) The English could be improved to more clearly express the research.
|
Yes |
Can be improved |
Must be improved |
Not applicable |
|
|
Does the introduction provide sufficient background and include all relevant references? |
(x) |
( ) |
( ) |
( ) |
|
Is the research design appropriate? |
(x) |
( ) |
( ) |
( ) |
|
Are the methods adequately described? |
(x) |
( ) |
( ) |
( ) |
|
Are the results clearly presented? |
(x) |
( ) |
( ) |
( ) |
|
Are the conclusions supported by the results? |
(x) |
( ) |
( ) |
( ) |
Comments and Suggestions for Authors
- A brief summary The main aim of this paper was to investigate whether children spent more or less time outdoors and the factors that influenced any changes.
- General concept comments
Article: The article was a well conceived and executed study that is well described, including limitations of the study and data. The conclusions are well supported by the results provided.
Review: The article positioned the research explicitly against other studies in this field. The article referred to most of the major literature available in the discussion topic relevant to the study. - Specific comments No inaccuracies could be found in areas such as table referencing, etc. Final manuscript will need to amend (Author, et al. 2016), remove sentence space in line 112 and 215 and ensure all references are formatted correctly. MDPI advise that this work will be done by them.
- The manuscript is clear, relevant for the field and presented in a well-structured manner
- The references were mostly recent and were appropriate.
- The manuscript is scientifically sound and the experimental design is appropriate to test the hypothesis but with acknowledged limitations of self-report design.
- A similar study could be constructed based on the methodological design and results reporting. The data is presented clearly and explicitly with appropriate selection of total data in relation to the research question.
- The conclusions consistent with the evidence and arguments are presented
- As the data and study was funded by a reputable government body and data collection carried out by a government sanctioned organisation it his highly unlikely that there are any ethical issues. However, an ethics approval number would further give confidence and and allow easier auditing.
Responses to the reviewer 1
Authors: Thank you very much for reading our paper and for the positive review report. We appreciate a lot that you like our paper and find it relevant for publishing. We have followed up your comments.
Authors: We have replaced Author et al. 2016 with Gundersen et al. 2016 all places and insert the correct reference in the list.
Authors: We have removed sentence space in line 112 and 215.
Authors: We have formatted the references in the text and in the list in accordance with the journal’s recommendations.
Authors: Regarding ethical considerations, we think the description in the chapter 2.4. Ethics approval is described in detail.

Reviewer 2 Report
Comments and Suggestions for Authors
Comments to authors
Thank you very much for an important paper on the rather sad story about the influence of Covid19 on the children’s outdoor play – despite the abundance of accessible green areas to most of the children. I can’t wait publishing more optimistic figures from my home countryJ
I think your study should be published, but I have – of course – some comments. Some overarching – some more about details
11 The time of the survey
I find it a little hard to grasp – is the study taking place during or after lock down. L. 105-131.
As I understand it the schools opened already summer 2020 and did not close again? (local measures are mentioned – what does that mean?). Did the practice in school etc. change? In some countries kindergartens, preschools and schools (afterschool care) changed practice to smaller groups and more teaching outside? Hence being more outside in these arenas, which might be important regarding the interpretation. When did organized activities start again? … You write that the “situation was largely normalized” at time of survey?
Another issue regarding timing. The survey was done in January – just after Christmas. It might influence answers a lot as outdoor frequency / time spent outside tends to vary over the year on the altitudes or N. At least this should be commented upon.
You are surveying on time “play and stay”. What is the role of transportation? Especially if children are roughly speaking in school most of the period? Are they walking/biking to school rather than taking public transport because of distance keeping?
What about weekend / weekday differences. Don’t know if that is something you have considered?
22 The tables
I think they are a little overwhelmingJ There are some repetition in table 3 (same figure mentioned in 1)
Table 2 look a little strange… is there something wrong in the Q What is the child’s and …..? The table should be made easier to read.
Table 4 and 5… could it be made more easy here to focus… you show a lot of insignificant numbers… maybe you should leave them out to make the table better support the text?
33 Age issues
In table 1, all the ages are lined up… but in the paper you only use children and adolescents. If that is the choice, the simplify table 1. But rather, I considered why you did not did the correlations on the full age span from 6 to 19? Have you a least tried that (The same with house hold income)
Figure 1… net change (L. 245)… would that be relevant to show for all ages… is there e.g. a dip and re bounce? Age 6 is very different from 12… Again, I think you should comment a little more on the choice of your two age groups.
Age of child and age of parents must – all other equal – be correlated. Hence, when you conclude on the age of parents it might just reflect age of child? I don’t recall that you comment on this.
4 Analysis
L. 212 What is meant by a cleaned data set
Did you consider weighting of the data? However, I think the survey sample seems quite representative. (don’t know much about Statistics Norway and possible biases of the panel).
Some of the studies that you quote have used modelling to test which variables may prescribe differences in being outdoor (in neighborhood and nature). Did you try other statistical methods than the ones you present?
55 Nature factor
I think it is interesting that the “outdoor reduction factor” (my expressionJ) is less regarding nature that outdoor areas in general. This could be interpreted as a relative increase of the nature component in the outdoor life… isn’t it worth commenting on? Yes, kids get less out in January 2021… but they choose nature relatively more often than they did before? It this something to build upon after all?
-
The points above may be of relevance for the discussion part also. You are very focused on organized leisure activities and their role for children’s time outside... You may underestimate the role of school/afterschool care and transport? But help the reader a bit on that point e.g. by addressing point 1 above.
I am looking forward to a published version of your paper
Author Response
Reviewer 2
Open Review
(x) I would not like to sign my review report
( ) I would like to sign my review report
Quality of English Language
(x) The quality of English does not limit my understanding of the research.
( ) The English could be improved to more clearly express the research.
|
Yes |
Can be improved |
Must be improved |
Not applicable |
|
|
Does the introduction provide sufficient background and include all relevant references? |
(x) |
( ) |
( ) |
( ) |
|
Is the research design appropriate? |
( ) |
(x) |
( ) |
( ) |
|
Are the methods adequately described? |
( ) |
(x) |
( ) |
( ) |
|
Are the results clearly presented? |
( ) |
(x) |
( ) |
( ) |
|
Are the conclusions supported by the results? |
(x) |
( ) |
( ) |
( ) |
Comments and Suggestions for Authors
Comments to authors
Thank you very much for an important paper on the rather sad story about the influence of Covid19 on the children’s outdoor play – despite the abundance of accessible green areas to most of the children. I can’t wait publishing more optimistic figures from my home countryJ
I think your study should be published, but I have – of course – some comments. Some overarching – some more about details
Authors: Thank you very much for your review report, which we find very relevant and helpful. Below are our responses to your comments.
11 The time of the survey
I find it a little hard to grasp – is the study taking place during or after lock down. L. 105-131.
Authors: The survey is taking place “in the period 12 to 30 January 2021” and after the lockdown (12 March 2020). It is a good point to clarify already in the introduction when our survey took part, and we added this in line 112-115 (highlighted): “During the second wave of COVID-19 infection, September 2020 to January 2021, national measures were replaced by local restrictions as needed and, in January 2021, by the time our survey was distributed, the situation for children and adolescents was largely normalized following a decreasing transmission trend.”
As I understand it the schools opened already summer 2020 and did not close again? (local measures are mentioned – what does that mean?). Did the practice in school etc. change? In some countries kindergartens, preschools and schools (afterschool care) changed practice to smaller groups and more teaching outside? Hence being more outside in these arenas, which might be important regarding the interpretation. When did organized activities start again? … You write that the “situation was largely normalized” at time of survey?
Authors: Yes, regarding changing adaptations during the schools reopening period, you have a good and relevant point for interpretation. We have now included this by adding this sentence in that section: “In the period with the reopening of preschools and schools, there were to some extent adaptations in teaching with smaller groups and also more teaching outdoors.”During the reopening period of preschools and schools, there were, to some extent, adaptations to teaching in smaller groups in addition to more teaching outdoors.”
Another issue regarding timing. The survey was done in January – just after Christmas. It might influence answers a lot as outdoor frequency / time spent outside tends to vary over the year on the altitudes or N. At least this should be commented upon.
Authors: Since the survey is asking retrospective, we suggest that low activity at the time the survey was sent out had no influence on the answers given. However, it is a good point and we added this sentence in the discussion under chapter 4.3. Strengths, limitations, and future directions: “A retrospective cross-sectional survey has obvious limitations but given the fast ap-pearance of the pandemic and the situation that parents and children were exposed to during it, we believe that the parents’ recollections’ on the situation before and after the pandemic are a relevant indication of children’s leisure behaviour. To overcome these limitations, we designed a methodology that involved asking for parents’ evaluations on children’s outdoor life without mention of COVID-19. This then gave us an indication of the main patterns of change that may be important to uncover in more detail in a cohort study. The survey questionnaire sent out on January 2021, referred to the children’s use of outdoor activities retrospectively during the COVID-19 pandemic. It is unlikely, therefore, that low outdoor activity among children in the time parents answered the questionnaire influenced their answer.”
You are surveying on time “play and stay”. What is the role of transportation? Especially if children are roughly speaking in school most of the period? Are they walking/biking to school rather than taking public transport because of distance keeping?
Authors: Walking/biking to school and organized activities is a part of what the parents reported as “play and stay”, meaning that “stay” includes use of nature in a broad sense. We think that this sentence/definition in the introduction includes situations when children walk or bike in nature to school from home: “Even studies conducted before the COVID-19 pandemic found that children were less likely to play and stay in their neighbourhood (within walking and cycling distance from home) and nature areas compared with children of a previous generation [11].”
What about weekend / weekday differences. Don’t know if that is something you have considered?
Authors: We have not consider to analyze differences between weekend-weekdays in the paper. We think it will add to much details, and want to save space.
22 The tables
I think they are a little overwhelmingJ There are some repetition in table 3 (same figure mentioned in 1)
Authors: Thanks. Yes it is the same parameters used in table 1 and table 3, but table 1 showed the definition/categories of the survey questionnaire, and table 3 sum up descriptive results. We may attached table 1 as an Supplementary file, however, we think both tables is necessary to understand the questionnaire. We have edited the layout of the tables.
Table 2 look a little strange… is there something wrong in the Q What is the child’s and …..? The table should be made easier to read.
Authors: We have simplified (e.g. removed letters) in the table 2 in accordance to your advice. Thanks.
Table 4 and 5… could it be made more easy here to focus… you show a lot of insignificant numbers… maybe you should leave them out to make the table better support the text?
Authors: Thanks. We have tried to simplify and edit the table 4 and table 5. In an earlier draft of the paper we had all the significant number in the text, but decide to leave them in table for readability of the paper. We have highlighted significant values with green to increase the readability of the paper.
33 Age issues
In table 1, all the ages are lined up… but in the paper you only use children and adolescents. If that is the choice, the simplify table 1. But rather, I considered why you did not did the correlations on the full age span from 6 to 19? Have you a least tried that (The same with house hold income)
Authors: Thanks. Table 1 is only testing representativity, so we decide to use all details regarding age classes that is included in the survey. Yes, we tested correlation for all age classes, but decide to use Point biserial correlations in the paper, using two main classes for different variables. We the same analyses as done in an comparable and relevant paper: Moore, S. A., Faulkner, G., Rhodes, R. E., Brussoni, M., Chulak-Bozzer, T., Ferguson, L. J., Mitra, R., O’Reilly, N., Spence, J. C., Vanderloo, L. M., & Tremblay, M. S. (2020). Impact of the COVID-19 virus outbreak on movement and play behaviours of Canadian children and youth: A national survey. International Journal of Behavioral Nutrition and Physical Activity, 17, Article 85. https://doi.org/10.1186/s12966-020-00987-8. Regarding age classes we decide to test differences between children (6-12 years) and teenagers (13-19 years), in regard to earlier published similar surveys: Skar, M., Wold, L.C., Gundersen, V. & O`Brien, L. 2016. Why do children not play in nearby nature? Results from a Norwegian survey. Journal of Adventure Education & Outdoor Learning 16(3): 239-255. DOI:10.1080/14729679.2016.1140587
Figure 1… net change (L. 245)… would that be relevant to show for all ages… is there e.g. a dip and re bounce? Age 6 is very different from 12… Again, I think you should comment a little more on the choice of your two age groups.
Authors: Here is a figure looking at those children/adolescent that play “less than usual”. We decide not to include age differences more in detailed in this paper. It is a question of scope and also paper length.
Figure. Differences between age classes distribution for those that play “less than usual” in neighbourhood and natural areas, and the age classes distribution from all sample.
Age of child and age of parents must – all other equal – be correlated. Hence, when you conclude on the age of parents it might just reflect age of child? I don’t recall that you comment on this.
Authors: Yes, of course. We put in a sentence in the discussion describing this: “Age of adolescents (13–19 years) and age of parents (40–80 years) is therefore correlated.”
4 Analysis
- 212 What is meant by a cleaned data set
Authors: We decide to delete cleaned in the first sentence, as the remove respondents as described in line 143-148. The most of the cleaning was done by us, first of all those respondents who don’t meet our target criteria and those that have incomplete answers. We edited this sentence: “Further data cleaning and verification was completed by us (e.g. target criteria, in-complete answers).”
Did you consider weighting of the data? However, I think the survey sample seems quite representative. (don’t know much about Statistics Norway and possible biases of the panel).
Authors: Yes, we tried weighting the data in different ways, and verify similar results. Yes, our respondents represented target population in a representative way.
Some of the studies that you quote have used modelling to test which variables may prescribe differences in being outdoor (in neighborhood and nature). Did you try other statistical methods than the ones you present?
Authors: No, we were inspired to do the same analyses as done in a very similar survey and paper: Moore, S. A., Faulkner, G., Rhodes, R. E., Brussoni, M., Chulak-Bozzer, T., Ferguson, L. J., Mitra, R., O’Reilly, N., Spence, J. C., Vanderloo, L. M., & Tremblay, M. S. (2020). Impact of the COVID-19 virus outbreak on movement and play behaviours of Canadian children and youth: A national survey. International Journal of Behavioral Nutrition and Physical Activity, 17, Article 85. https://doi.org/10.1186/s12966-020-00987-8.
55 Nature factor
I think it is interesting that the “outdoor reduction factor” (my expressionJ) is less regarding nature that outdoor areas in general. This could be interpreted as a relative increase of the nature component in the outdoor life… isn’t it worth commenting on? Yes, kids get less out in January 2021… but they choose nature relatively more often than they did before? It this something to build upon after all?
-
The points above may be of relevance for the discussion part also. You are very focused on organized leisure activities and their role for children’s time outside... You may underestimate the role of school/afterschool care and transport? But help the reader a bit on that point e.g. by addressing point 1 above.
Authors: Good point. We read through the discussion and add some words to underline your point. We are not asking about the situation for outdoor play in January, but asking retrospective the play and stay outside the last ten month, compare to similar situation before the lockdown.
I am looking forward to a published version of your paper
Authors: Thank you for very relevant comments that helps improve our paper.

Reviewer 3 Report
Comments and Suggestions for Authors
Reviewer’s report:
I think this paper is fine; It does add to the repository of knowledge of importance of healthy and social development in childhood adolescence.
The authors seem to be fond of using e.g in citing refs. I don’t think this is the norm and they need to remove this e.g. from many places they have used this in lines such as 27,37,40,41, 47,48,72,99,178,209 and so on. Read your paper carefully and remove these as they look odd.
Some sentences (lines 271 to 293) are quite long and difficult to follow. For example: "Adolescents (13–19 years) spent more time on screens and social media (0.17) than the children (6–12 years), and these results was supported by the finding that older parents (40–80 years) had adolescents who spent more time on screens and social media than younger parents (20–39 years) (0.10)." This could be broken down into smaller, more digestible parts. Additionally, the use of values near the threshold (e.g., 0.10) are very weak correlations, and this should be emphasized more clearly. Instead the authors have selected to insert these in here without much information or clarity. There really needs to be a series of smaller coherent sentences which should come under subheadings where applicable (demographic, social and environmental) rather than all being too long and overwhelming paragraphs. This will add clarity for both the authors and the readers.
The transition between ideas can be smoother. For example, the jump from screen time to family structure to outdoor activities feels abrupt. Use clearer transitions to guide the reader through the associations.
Also, there’s a typo in line 275 - "these results was supported" (should be "were supported").
Tables 4 and 5 are rather overwhelming numerically to follow- is there a graphical way of representation of your data?
Lines 394-399 are rather obtuse. What are the authors trying to state here? Can this be simplified?
Lines 410 -422: The statement "only 7% of the parents reported that their children or adolescent did not participate in any kind of organized leisure activities" is clear, but it lacks context. Is this number high or low compared to pre-pandemic levels or other countries? Providing a comparison would give readers a better sense of its significance.
Lines 415-416: "Many parents stated that their child did not start a new hobby/activity during the pandemic" is presented without quantification. How many is "many"? Providing specific data here would make the argument stronger.
Error in grammar in places : lines 418-420: The sentence “parents’ motivation to engage in recreational activities with their child for their children to engage in outdoor play without supervision or is crucial for outdoor activity levels” is grammatically awkward and unclear. It could be rewritten for clarity, such as: “Parents’ motivation to engage in recreational activities with their children is crucial for encouraging outdoor play, especially unsupervised play.”
Overall grammar needs to be improved to a better standard; sentences are too long; the citations using e.g seems unnecessary; many typos in paper.
I was at a loss to understand the removal of ref by “author” why? Was it removed?
Comments on the Quality of English Language
The authors really need to use subheadings to help them and the readers follow their methods and results and discussions with clarity, since their quality of English is lost due to very long sentences. There are many typos and this needs to be corrected. I have identified a few.
Author Response
Reviewer 3
Open Review
(x) I would not like to sign my review report
( ) I would like to sign my review report
Quality of English Language
( ) The quality of English does not limit my understanding of the research.
(x) The English could be improved to more clearly express the research.
|
Yes |
Can be improved |
Must be improved |
Not applicable |
|
|
Does the introduction provide sufficient background and include all relevant references? |
(x) |
( ) |
( ) |
( ) |
|
Is the research design appropriate? |
(x) |
( ) |
( ) |
( ) |
|
Are the methods adequately described? |
( ) |
( ) |
(x) |
( ) |
|
Are the results clearly presented? |
( ) |
( ) |
(x) |
( ) |
|
Are the conclusions supported by the results? |
( ) |
( ) |
(x) |
( ) |
Comments and Suggestions for Authors
Reviewer’s report:
I think this paper is fine; It does add to the repository of knowledge of importance of healthy and social development in childhood adolescence.
Authors: Thank you for reviewing our paper and thank you for good and relevant feedback.
The authors seem to be fond of using e.g in citing refs. I don’t think this is the norm and they need to remove this e.g. from many places they have used this in lines such as 27,37,40,41, 47,48,72,99,178,209 and so on. Read your paper carefully and remove these as they look odd.
Authors: We have edited these references in accordance with the journal reference style.
Some sentences (lines 271 to 293) are quite long and difficult to follow. For example: "Adolescents (13–19 years) spent more time on screens and social media (0.17) than the children (6–12 years), and these results was supported by the finding that older parents (40–80 years) had adolescents who spent more time on screens and social media than younger parents (20–39 years) (0.10)." This could be broken down into smaller, more digestible parts. Additionally, the use of values near the threshold (e.g., 0.10) are very weak correlations, and this should be emphasized more clearly. Instead the authors have selected to insert these in here without much information or clarity. There really needs to be a series of smaller coherent sentences which should come under subheadings where applicable (demographic, social and environmental) rather than all being too long and overwhelming paragraphs. This will add clarity for both the authors and the readers.
The transition between ideas can be smoother. For example, the jump from screen time to family structure to outdoor activities feels abrupt. Use clearer transitions to guide the reader through the associations.
Authors: Thanks. We have looked carefully through the paper and simplify and shortened sentences. A skilled language editor checked the whole manuscript once again, and we attached all the changes (incl. track changes) to the Editor og the journal.
Also, there’s a typo in line 275 - "these results was supported" (should be "were supported").
Authors: Done
Tables 4 and 5 are rather overwhelming numerically to follow- is there a graphical way of representation of your data?
Authors: We have been thinking about other way to present these results better. We presented the data as done in a very similar study, carried out by Moore, S. A., Faulkner, G., Rhodes, R. E., Brussoni, M., Chulak-Bozzer, T., Ferguson, L. J., Mitra, R., O’Reilly, N., Spence, J. C., Vanderloo, L. M., & Tremblay, M. S. (2020). Impact of the COVID-19 virus outbreak on movement and play behaviours of Canadian children and youth: A national survey. International Journal of Behavioral Nutrition and Physical Activity, 17, Article 85. https://doi.org/10.1186/s12966-020-00987-8.
Lines 394-399 are rather obtuse. What are the authors trying to state here? Can this be simplified?
Authors: Thanks. We have rewritten and shortened these two sentences to make them clearer: “The situation in ‘green’ Norway seems to be much the same as in other Western countries, where spending time outdoors in natural settings may be at risk of being forgot-ten among the scheduled targets and competing indoor screen activities (e.g. [9,10,53]). Playing games also include landscapes and natural scenery on the screen, but these settings are perfectly arranged for the game activity and cannot replace physical nature experiences [55].”
Lines 410 -422: The statement "only 7% of the parents reported that their children or adolescent did not participate in any kind of organized leisure activities" is clear, but it lacks context. Is this number high or low compared to pre-pandemic levels or other countries? Providing a comparison would give readers a better sense of its significance.
Authors: Good point. We added this to the sentence: “Our data confirmed the importance of organized activities for children’s life out-doors in Norway compared with other Western countries (e.g. [24]), where only 7% of the parents reported that their children or adolescents did not participate in any kind of organized leisure activities.”
Lines 415-416: "Many parents stated that their child did not start a new hobby/activity during the pandemic" is presented without quantification. How many is "many"? Providing specific data here would make the argument stronger.
Authors: Good point. We rewrote the sentence: “The statement that their child did not start a new hobby/activity during the pandemic got a low score (mean: 2.49). This can be interpreted as meaning that there were few opportunities for children to explore new arenas during the pandemic, and this was especially the case for adolescents.”
Error in grammar in places : lines 418-420: The sentence “parents’ motivation to engage in recreational activities with their child for their children to engage in outdoor play without supervision or is crucial for outdoor activity levels” is grammatically awkward and unclear. It could be rewritten for clarity, such as: “Parents’ motivation to engage in recreational activities with their children is crucial for encouraging outdoor play, especially unsupervised play.”
Authors: Thanks. We edited the sentence in accordance with your advice: “We noted that parents considered that their child had less busy weekdays and spent more time with adults, and these findings underline that parents’ motivation to engage in recreational activities with their children is crucial for encouraging outdoor play, especially unsupervised play [12,16,24].”
Overall grammar needs to be improved to a better standard; sentences are too long; the citations using e.g seems unnecessary; many typos in paper.
Authors: We have checked the whole text once again with a skilled language editor, and attached all changes (incl. track changes) to the Editor of the journal.
I was at a loss to understand the removal of ref by “author” why? Was it removed?
Authors: Yes, we have removed this.
Comments on the Quality of English Language
The authors really need to use subheadings to help them and the readers follow their methods and results and discussions with clarity, since their quality of English is lost due to very long sentences. There are many typos and this needs to be corrected. I have identified a few.
Authors: Good point. We have looked carefully through the text to increase the readability. Additional a skilled language editor checked the text.
